# A Consistent Flow Model Learning Both Where to Go and How to Move

## Abstract

Under the framework of rectified flow, generative neural networks are trained to a single vector field specified by straight paths between data samples and random samples from the prior. This work reveals equivalent two forms of the learning problem, with each form setting up a fitting target for the neural network. We then introduce a new model, BiFlow, which is trained with two complementary targets-the velocity of the vector field and the likely destination of ODE paths-thereby endows the model with *local* sensitivity (how to move now) and *global* awareness (where the path should end). The new design uses a single head with a binary mode flag to output either prediction, plus a lightweight consistency loss that ties them together. It drops into existing pipelines: no architectural overhaul, the usual conditioning, therefore adding no extra cost to the generation procedure. Our experiment shows that the new design stabilizes optimization and improves the straightness of generation paths. On two image generation tasks, BiFlow substantially improves generation quality over rectified flow.

## 1 Introduction

Generative models based on ordinary differential equations (ODEs) (Lipman et al., 2022), particularly rectified flows (Liu et al., 2022), have emerged as a powerful alternative to traditional diffusion models, enabling high-quality synthesis with significantly fewer sampling steps. The neural network learns the vector field of an ODE that transforms a simple prior distribution into the data distribution.

In rectified flow, the neural network is trained to match the vectors defined by straight paths between training examples and random samples from the prior distribution. With the simple form of the straight paths, we discover a new form of learning objective for training the neural network, which directly learns to predict a data sample from its corrupted version. Compared with rectified flow, where the neural network predicts the instantaneous update of a sample (the velocity) along the ODE path, the network in our new form directly predicts the likely destination of the ODE path.

In this work, we combine the new learning form with flow matching and introduce **BiFlow**, a framework that trains a single model to simultaneously predict two complementary targets: the instantaneous velocity and the likely destination of the ODE path. Our key insight is that the neural network receives training signals in two forms and thus gains extra power in learning the vector field.

Our approach is based on a simple and efficient architecture design. BiFlow uses a single network backbone to compute the two fitting targets, with a binary mode flag to indicate the type of the target. The design adds negligible overhead to a baseline rectified flow model and incurs the same level of inference costs in the generation procedure. At sampling time, we leverage both learned predictors with a theoretically-grounded, time-scheduled switching policy that uses the more reliable estimate at each stage of the ODE solve. Experiments on ImageNet show that BiFlow substantially outperforms the rectified flow baseline, demonstrating the power of learning both where to go and how to move.

To unpack where the gains come from under our *single-model, multi-goal* design, we run focused ablations: (i) train single-goal variants (velocity-only and destination-only) to isolate the effect of supervising multiple targets within one trunk, and (ii) turn off the consistency loss to test whether algebraically tying the two predictions is necessary. Across datasets and backbones, the single-goal variants narrow but do not close the gap, while removing the consistency term noticeably degrades

FID and path coherence, indicating that *joint supervision in one model* together with an *explicit consistency loss* is essential to BiFlow's improvements.

Our contributions can be summarized as follows:

- **Dual-Target Framework:** We propose BiFlow, a novel training framework for rectified flows that supervises a single model on two complementary targets: the instantaneous velocity and the likely destination.

- **Consistency Loss:** We introduce a consistency loss that enforces the analytical relationship between the two predictions. This improves optimization and ensures the learned vector fields are coherent.

- **Efficient Implementation:** Our method uses a single, shared-head network with a binary mode flag, resulting in no architectural changes, minimal training overhead, and identical sampling cost (one network evaluation per ODE step) compared to standard rectified flows.

- **Strong Empirical Gains:** Empirically, BiFlow substantially outperforms the standard rectified flow baseline on ImageNet $256 \times 256$, achieving significant FID improvements with the same sampling cost and negligible additional training overhead.

## 2 RELATED WORKS

**Diffusion models and probability-flow ODEs.** Denoising diffusion models and score-based generative modeling established the modern baseline for high-fidelity synthesis, sampling either reverse SDEs or their probability-flow ODEs (Ho et al., 2020; Song et al., 2020). Subsequent work improved objectives, parameterizations, and samplers (Nichol & Dhariwal, 2021; Karras et al., 2022), and DiT scaled diffusion backbones with transformers (Peebles & Xie, 2023).

**Flow matching and rectified flows.** Flow Matching trains ODE vector fields by regressing the Bayes velocity along reference paths (Lipman et al., 2022). Rectified Flow learns nearly straight trajectories and popularized "ReFlow" straightening and one-/few-step generation (Liu et al., 2022). Recent papers refine training and straightening—e.g., improved RF with minimal ReFlow rounds (Lee et al., 2024), one-step text-to-image distillation via RF (Liu et al., 2023), and optimal-transport straightness in a single FM step (Kornilov et al., 2024).

**Interpolants and transformer backbones.** Beyond diffusion-specific paths, stochastic interpolants unify flows and diffusions and motivate scalable transformer implementations (Albergo et al., 2023) ; building NFs with interpolants (Albergo & Vanden-Eijnden, 2022). SiT (Ma et al., 2024) has been treated as a benchmark to combine flow models and DiT backbones for image negeration.

**Closest to BiFlow: dual objectives and velocity–endpoint links.** Several recent papers pursue the idea of combining local and global training signals but differ in mechanism. *Flow-Anchored Consistency Models* (FACM) (Peng et al., 2025) jointly train a consistency model and a flow-matching task, using conditioning to toggle tasks, yet they do not impose an explicit algebraic coupling between the two predictions. *Consistency Flow Matching* (Yang et al., 2024) regularizes only the velocity field by enforcing self-consistency across start times to encourage straight trajectories, without learning a separate endpoint predictor. *MeanFlow* (Geng et al., 2025) learns an average (time-integrated) velocity and provides an identity linking this average to the instantaneous one—conceptually close to making two predictions agree, but implemented via a different target rather than two coordinated outputs with a constraint. Outside the flow-matching family, diffusion work has explored multiple parameterizations and dual outputs—for example, predicting both noise and the clean sample, sometimes with a learned mixing strategy—which validates multi-output training but remains diffusion-specific (e.g., Benny & Wolf (2022) ). In contrast, BiFlow uses one shared backbone with a binary control bit to produce either a local motion prediction or a global endpoint prediction, and it penalizes disagreement between them, enabling compute-parity switching at inference without distillation or extra forward passes.

## 3 BACKGROUND

**Flow matching.** Flow matching aims to model data drawn from an unknown distribution $p_*(\mathbf{x})$ on $\mathbb{R}^d$. A flow-matching model specifies an ODE on the random variable $\mathbf{x}$ that transports samples from a standard distribution $p_0(\mathbf{x})$ to a target distribution $p_1(\mathbf{x})$. Typically, $p_0(\mathbf{x})$ is a standard Gaussian, and $p_1(\mathbf{x})$ approximates the data distribution.

The ODE is defined by a time-dependent vector field $\mathbf{v}_\theta(\mathbf{x}_t, t) : \mathbb{R}^d \times [0,1] \to \mathbb{R}^d$, so that $\mathrm{d}\mathbf{x}_t = \mathbf{v}_\theta(\mathbf{x}_t, t)\,\mathrm{d}t$ with $\mathbf{x}_0 \sim p_0$. The vector field $\mathbf{v}_\theta$ is parameterized by a neural network. The learning objective is to approximate a vector field $u_*(\mathbf{x}, t)$ that transports the standard distribution $p_0$ to the target distribution $p_*$.

Without knowing $u_*(\mathbf{x}, t)$, flow matching (Lipman et al., 2022) trains $\mathbf{v}_\theta$ by regressing it toward a conditional vector field. Let $c(\mathbf{x}_0, \mathbf{x}_1)$ be a coupling of $p_0$ and $p_*$, that is, a joint distribution whose marginals are $p_0$ and $p_*$. Drawing $(\mathbf{x}_0, \mathbf{x}_1) \sim c$, the network then needs only to match the velocity along a path connecting each pair $(\mathbf{x}_0, \mathbf{x}_1)$ (Tong et al., 2023b).

**Rectified Flow Matching.** Rectified flow is a special case of flow matching: the coupling is $p(\mathbf{x}_0, \mathbf{x}_1) = p_0(\mathbf{x}_0)p_*(\mathbf{x}_1)$, and the path between the pair $(\mathbf{x}_0, \mathbf{x}_1)$ is a straight line. Formally, for any $t \in [0,1]$, the conditional vector field is:

$$\mathbf{u}(\mathbf{x}_t, t|\mathbf{x}_0, \mathbf{x}_1) := \mathbf{x}_1 - \mathbf{x}_0, \qquad \mathbf{x}_t = (1-t)\mathbf{x}_0 + t\mathbf{x}_1$$

The training objective for the neural vector field $\mathbf{v}_\theta(\mathbf{x}_t, t)$ is:

$$L_{\mathrm{FM}}(\theta) \;=\; \mathbb{E}_{c(\mathbf{x}_0, \mathbf{x}_1)}\Big[\, \big\| \mathbf{v}_\theta(\mathbf{x}_t, t) - \mathbf{u}(\mathbf{x}_t, t|\mathbf{x}_0, \mathbf{x}_1) \big\|_2^2 \,\Big], \tag{1}$$

The objective is minimized by $\mathbf{v}_\theta(\mathbf{x}_t, t) = \mathbb{E}_{c(\mathbf{x}_0, \mathbf{x}_1)}[\mathbf{u}(\mathbf{x}_t, t|\mathbf{x}_0, \mathbf{x}_1)]$, which transports $p_0$ to the data distribution $p_*$.

**Sampling.** Given a trained $\mathbf{v}_\theta$, we can draw new samples by integrating the ODE path starting from a random $\mathbf{x}_0 \sim p_0$.

$$\mathbf{x}_1 \;=\; \mathbf{x}_0 + \int_{t=0}^{1} \mathbf{v}_\theta(\mathbf{x}_t, t)\mathrm{d}t, \qquad x_{t=0} \sim p_0. \tag{2}$$

A numerical ODE solver (e.g., Dopri5 or fixed-step Euler/Heun) advances $x_t$ from $t = 0$ to $t = 1$, yielding a sample from $p_1$, which should approximate the data distribution $p_*$.

## 4 BIFLOW MODEL

**Motivation.** BiFlow is motivated by re-examining the flow-matching objective. For a $t$ and $\mathbf{x}_t$ pair, the regression target for $\mathbf{v}_\theta(\mathbf{x}_t, t)$ depends only on those pairs $(\mathbf{x}_0, \mathbf{x}_1)$ that are consistent with $\mathbf{x}_t$; consequently, the pairs used to optimize $\mathbf{v}_\theta(\mathbf{x}_t, t)$ are restricted to the set

$$S(\mathbf{x}_t, t) = \{(\mathbf{x}_0, \mathbf{x}_1) : \mathbf{x}_t = (1-t)\mathbf{x}_0 + t\mathbf{x}_1\}.$$

With the relation $\mathbf{x}_0 = \frac{\mathbf{x}_t - t\mathbf{x}_1}{1-t}$ and by restricting the coupling $c(\mathbf{x}_0, \mathbf{x}_1)$ to the feasible set $\mathcal{S}(\mathbf{x}_t, t)$, we obtain $\tilde{c}(\mathbf{x}_1) \propto c(\mathbf{x}_0, \mathbf{x}_1)$ on the set $S(\mathbf{x}_t, t)$. Then for $t \in (0, 1)$, the fitting target becomes:

$$\mathbb{E}_c\left[\|\mathbf{v}_\theta(\mathbf{x}_t, t) - (\mathbf{x}_1 - \mathbf{x}_0)\|_2^2\right] = \mathbb{E}_{\tilde{c}}\left[\left\|\mathbf{v}_\theta(\mathbf{x}_t, t) - \frac{\mathbf{x}_1 - \mathbf{x}_t}{1-t}\right\|_2^2\right]$$

$$= \frac{1}{(1-t)^2}\mathbb{E}_{\tilde{c}}\left[\|((1-t)\mathbf{v}_\theta(\mathbf{x}_t, t) + \mathbf{x}_t) - \mathbf{x}_1\|_2^2\right]. \tag{3}$$

Since $t$ is sampled from a continuous distribution (e.g., $\mathcal{U}(0,1)$), the endpoints $t \in \{0, 1\}$ have probability zero and can be ignored. From the form above, the optimal value for $\mathbf{v}_\theta$ satisfies

$$((1-t)\mathbf{v}_\theta(\mathbf{x}_t, t) + \mathbf{x}_t) = \mathbb{E}_{\tilde{c}}[\mathbf{x}_1].$$

Noting that the left-hand side only involves $(\mathbf{x}_t, t)$, we can define a new function $\mathbf{m}_\theta(\mathbf{x}_t, t)$ to replace the left-hand side. We then train by directly regressing $\mathbf{m}_\theta(\mathbf{x}_t, t)$ toward the data endpoint $\mathbf{x}_1$. The relationship between $\mathbf{m}_\theta$ and $\mathbf{v}_\theta$ is:

$$\mathbf{m}_\theta = (1-t)\mathbf{v}_\theta(\mathbf{x}_t, t) + \mathbf{x}_t, \qquad \mathbf{v}_\theta = \frac{\mathbf{m}_\theta(\mathbf{x}_t, t) - \mathbf{x}_t}{1-t} \tag{4}$$

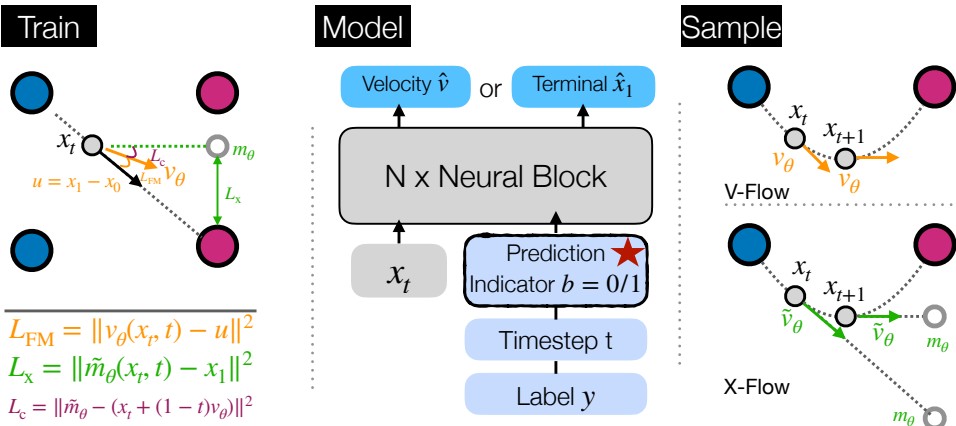

Figure 1: **BiFlow overview: (A) Training.** Linear rectified path $x_t = (1 - t)x_0 + tx_1$ with instantaneous velocity $u = x_1 - x_0$; Training minimizes $L_{\text{FM}} = \|\mathbf{v}_\theta - u\|^2$, $L_{\text{x}} = \|\tilde{\mathbf{m}}_\theta - x_1\|^2$, and the consistency term $L_{\text{c}} = \|\tilde{\mathbf{m}}_\theta - (x_t + (1 - t)\mathbf{v}_\theta)\|^2$. **(B) Model.** Single network $\text{nn}_\theta(x_t, t, y, b)$ with summed embeddings $E_t(t) + E_y(y) + E_b(b)$; mode $b{=}0$ outputs $\mathbf{v}_\theta$ (velocity) and $b{=}1$ outputs $\tilde{\mathbf{m}}_\theta$ (endpoint). **(C) Sampling.** Sampling integrates the probability-flow ODE with a per-step drift chosen from $\mathbf{v}_\theta$ and $(\tilde{\mathbf{m}}_\theta - x_t)/(1 - t)$ (Equation (9)); any selection/blending policy may be used without retraining.

**The BiFlow method.** We introduce another neural function $\tilde{\mathbf{m}}_\theta$ to directly learn $\mathbf{x}_1$. The learning objective for $\mathbf{m}_\theta$ is:

$$L_{\text{x}}(\theta) = \|\tilde{\mathbf{m}}_\theta(\mathbf{x}_t, t) - \mathbf{x}_1\|_2^2 \tag{5}$$

Here we use $\theta$ to denote the network parameters, since later both $\mathbf{v}_\theta$ and $\tilde{\mathbf{m}}_\theta$ are parameterized by the same network. Compared with equation 3, this objective removes the leading factor $1/t^2$ to improve training stability; empirically, we find that this change has negligible impact on performance.

Since $\mathbf{v}_\theta$ and $\tilde{\mathbf{m}}_\theta$ are computed from different neural functions, we need to make them consistent with each other. We include another consistency term in the training objective:

$$L_{\text{c}} = \|\tilde{\mathbf{m}}_\theta(\mathbf{x}_t, t) - \mathbf{m}_\theta(\mathbf{x}_t, t)\|_2^2 = \|\tilde{\mathbf{m}}_\theta(\mathbf{x}_t, t) - ((1 - t)\mathbf{v}_\theta(\mathbf{x}_t, t) + \mathbf{x}_t)\|_2^2. \tag{6}$$

The entire training objective is:

$$\mathcal{L}(\theta) = w_v L_{\text{FM}}(\theta) + w_x L_x(\theta) + w_c L_c(\theta) \tag{7}$$

**Analysis.** In theory, $\mathbf{v}_\theta$ and $\tilde{\mathbf{m}}_\theta$ are mathematically equivalent: with infinite data and model capacity, both attain the same population optimum and satisfy equation 4. In that idealized limit, the auxiliary parameterization is redundant. In practice, however, data are limited and optimization is noisy. At each iteration, the training signal for $\mathbf{v}_\theta(\mathbf{x}_t, t)$ is $(\mathbf{x}_1 - \mathbf{x}_0)$, a noisy surrogate of the endpoint $\mathbf{x}_1$, whereas $\tilde{\mathbf{m}}_\theta(\mathbf{x}_t, t)$ is trained directly against the clean target $\mathbf{x}_1$. Thus the two neural functions receive different supervision. Adding a consistency term couples them, so that $\mathbf{v}_\theta$ and $\tilde{\mathbf{m}}_\theta$ inform each other and better fit the data distribution.

**Parameterization.** While $\mathbf{v}_\theta$ and $\tilde{\mathbf{m}}$ play different roles, their required information is similar. In this work, we train a *single* network $\text{nn}_\theta$ to compute both functions. In particular, we use a binary flag $b \in \{0, 1\}$ to indicate which function it computes:

$$\mathbf{v}_\theta(\mathbf{x}_t, t) := \text{nn}_\theta(\mathbf{x}_t, t, b{=}0), \qquad \tilde{\mathbf{m}}_\theta(\mathbf{x}_t, t) := \text{nn}_\theta(\mathbf{x}_t, t, b{=}1). \tag{8}$$

From $\tilde{\mathbf{m}}$, we derive another flow:

$$\tilde{\mathbf{v}}_\theta = \frac{\tilde{\mathbf{m}}_\theta(\mathbf{x}_t, t) - \mathbf{x}_t}{1 - t} \tag{9}$$

We refer to the original flow $\mathbf{v}_\theta$ as **V-Flow** and to our new flow $\tilde{\mathbf{v}}_\theta$ as **X-Flow**.

We also consider conditional generation. Let $y$ denote a conditioning variable (e.g., a class label). We augment the inputs with $y$, so the network becomes $\text{nn}_\theta(\mathbf{x}_t, t, b, y)$.

In the implementation, we embed the time $t$, the indicator $b$, and the condition $y$ with vectors and send their sum into the neural network, $e(t, y, b) = e_t(t) + e_y(y) + e_b(b)$. Here we assume $y$ is a condition with a simple form.

**Sampling.** In the generative procedure, we have two vector fields $\mathbf{v}_\theta(\mathbf{x}_t, t)$ and $\tilde{\mathbf{v}}_\theta(\mathbf{x}_t, t)$ for sampling. We can use either of them for sampling with path integration. We can also blend them in the sampling procedure. Our experiment later shows that the $\text{ta}\mathbf{v}_\theta(\mathbf{x}_t, t)$ and $\tilde{\mathbf{v}}$ do not have significant differences.

Below we show a mixed sampling procedure that switches from $\tilde{\mathbf{v}}_\theta$ to the $\mathbf{v}_\theta$ in the middle:

$$\dot{\mathbf{x}}_t = \begin{cases} \tilde{\mathbf{v}}_\theta(\mathbf{x}_t, t), & t \leq \tau, \\ \mathbf{v}_\theta(\mathbf{x}_t, t), & t > \tau, \end{cases} \tag{10}$$

In our experiment, we use $\tau = 0.5$.

**Computation.** Compared with a baseline that predicts $\mathbf{v}_\theta$ with a single network, our model introduces only a minimal number of additional parameters –the embedding of the binary switch $b$. Training time increases only slightly because the computations for $\mathbf{v}_\theta$ and $\mathbf{m}_\theta$ are largely shared. At sampling time, the runtime is nearly identical to that of the baseline flow-matching model: the compute budget is essentially unchanged, and we simply toggle $b$ to select $\mathbf{v}_\theta$ or $\tilde{\mathbf{v}}_\theta$.

| Algorithm 1: BiFlow training (single head) | Algorithm 2: BiFlow sampling (ODE) |
|---|---|
| 1: **Input:** $(w_v, w_x, w_c)$, clamp $\varepsilon$, optimizer | 1: **Input:** prior $x_0 \sim p_0$, condition $y$, clamp $\varepsilon$ |
| 2: **for** minibatches $(x_0, x_1, y)$ in dataset **do** | 2: Choose per-step drift rule using equation 10 |
| 3:     Draw $t \sim \mathcal{U}([\varepsilon, 1-\varepsilon])$ | 3: **for** $t$ from 0 to 1 with an ODE solver **do** |
| 4:     $x_t \leftarrow (1-t)x_0 + tx_1, \quad u \leftarrow x_1 - x_0$ | 4:     $\mathbf{v}_\theta \leftarrow \text{nn}_\theta(x_t, t, y, b{=}0)$ |
| 5:     $\mathbf{v}_\theta \leftarrow \text{nn}_\theta(x_t, t, y, b{=}0)$ | 5:     $\hat{\mathbf{m}}_\theta \leftarrow \text{nn}_\theta(x_t, t, y, b{=}1)$ |
| 6:     $\hat{\mathbf{m}}_\theta \leftarrow \text{nn}_\theta(x_t, t, y, b{=}1)$ | 6:     $\tilde{\mathbf{v}}_\theta \leftarrow (\hat{x}_1 - x_t)/\max(1-t, \varepsilon)$ |
| 7:     Compute $L_{\text{FM}}, L_{\text{x}}, L_{\text{c}}$ | 7:     $\dot{x}_t \leftarrow \text{mix}(\mathbf{v}_\theta, \tilde{\mathbf{v}}_\theta, t)$ |
| 8:     $\mathcal{L}(\theta) \leftarrow w_v L_{\text{FM}}(\theta) + w_x L_x(\theta) + w_c L_c(\theta)$ | 8:     Advance $x_t$ one ODE step using $\dot{x}_t$ |
| 9:     Update $\theta$ with $\nabla_\theta L$ | 9: **end for** |
| 10: **end for** | 10: **return** $x_{t=1}$ |

## 5 EXPERIMENTS

We empirically evaluate BiFlow on image generation tasks. We first check the performance of BiFlow and then examine the effect of the proposed training method with a series of ablation studies.

**Datasets and evaluation metrics.** The experiment is conducted on the two well-known datasets, CIFAR-10 (Krizhevsky, 2009) and ImageNet $256 \times 256$ (Deng et al., 2009). The evaluation metrics are Fréchet Inception Distance (FID) (Heusel et al., 2017), sFID (Siarohin et al., 2019), Inception Score (IS) (Salimans et al., 2016), and Precision (Kynkäänniemi et al., 2019). Lower FID/sFID and higher IS and Precision correspond to better performance.

Our primary baseline for performance comparison is Rectified Flow. All methods share the same neural backbone architecture, with the only difference being the additional switch $b$ introduced in BiFlow. BiFlow supports three sampling strategies: V-Flow, X-Flow, and a mixed approach defined in equation 10. We denote the three strategies with suffixes -V, -X, and -Mix, respectively.

### 5.1 EVALUATION WITH CIFAR-10

For our CIFAR-10 experiments, we use a U-Net backbone with an architecture similar to the one used in DDPM++ (Nichol & Dhariwal, 2021), which is a standard for this dataset. Both the baseline and BiFlow models are trained for 250K iterations. For evaluation, we generate samples using Heun's method with 250 steps to ensure stable ODE integrations.

As shown in Table 1, BiFlow demonstrates a significant performance improvement over the standard rectified flow baseline on CIFAR-10. The BiFlow-Mix and BiFlow-X variants achieve the best FID

Table 1: Main results on CIFAR-10 (250K iterations, 250 sampling steps). BiFlow consistently outperforms the Rectified Flow baseline.

| Model | FID↓ | sFID↓ | IS↑ | Precision↑ |
|---|---|---|---|---|
| RectifiedFlow | 3.12 | 3.34 | 9.47 | 0.72 |
| BiFlow-V | 2.49 | 2.93 | 9.61 | 0.73 |
| BiFlow-X | **2.41** | 2.88 | 9.64 | 0.73 |
| BiFlow-Mix ($\tau{=}0.5$) | **2.41** | **2.83** | **0.65** | 0.73 |

score of **2.41**, a substantial 22.8% relative improvement over the baseline's 3.12. All BiFlow sampling modes achieve better performance than the baseline method, indicating that the new training scheme also improves the V-flow $\mathbf{v}_\theta$. This strong performance with a convolutional U-Net architecture (Ronneberger et al., 2015) (as opposed to a Transformer (Vaswani et al., 2017)) highlights the general applicability and benefit of our proposed method.

## 5.2 EVALUATION WITH IMAGENET 256x256

Our experiments on ImageNet use the DiT-L/4 and DiT-XL/2 (Peebles & Xie, 2023) backbones as the foundation for our models. Here L and XL are mode sizes defined in the original paper. The number are patch sizes used by the generative model. We use "-L" and "-XL" to indcate these two settings.

Our standard Rectified Flow baselines are trained following the SiT (Ma et al., 2024). The BiFlow model employs the same backbone architecture, except for the input of the binary switch $b$. The L/4 models are trained for 500K iterations and the XL/2 models for 800K iterations, both with a constant learning rate of $1 \times 10^{-4}$. For evaluation, all models are sampled using the Dopri5 ODE solver (Dormand & Prince, 1980) with 250 steps.

In this experiment, we include DiT as a baseline. DiT uses the same network backbone but is a diffusion-based generative method. We also train a flow matching model with only X-flow: we train $\tilde{\mathbf{m}}_\theta$ with the objective $L_x$ and generate images with the derived X-flow $\tilde{\mathbf{v}}_\theta$. We denote this method as RectifiedFlow-L-X or RectifiedFlow-XL-X. All experiments are conducted with classifier free guidance (cfg) (Ho & Salimans, 2022) during sampling.

Table 2: Performances on ImageNet $256 \times 256$. BiFlow with all three sampling methods has significant improvements over the baseline model.

(a) L/4 (cfg=4.0, 500K training steps, 250 sampling steps)

| Model | FID↓ | sFID↓ | IS↑ | Precision↑ |
|---|---|---|---|---|
| DiT-L | 11.84 | 12.22 | 116.50 | 0.63 |
| RectifiedFlow-L | 11.53 | 12.06 | 110.75 | 0.65 |
| RectifiedFlow-L-X | 11.48 | 12.02 | 112.63 | 0.65 |
| BiFlow-L-V | 10.63 | 11.90 | 137.86 | 0.67 |
| BiFlow-L-X | 9.92 | 10.04 | 143.09 | **0.68** |
| BiFlow-L-Mix ($\tau{=}0.5$) | **9.85** | **10.02** | **145.61** | **0.68** |

(b) XL/2 (cfg=1.5, 800K training steps, 250 sampling steps)

| Model | FID↓ | sFID↓ | IS↑ | Precision↑ |
|---|---|---|---|---|
| DiT-XL | 6.41 | 6.67 | 175.01 | 0.73 |
| RectifiedFlow-XL | 6.22 | 6.65 | 157.46 | 0.73 |
| RectifiedFlow-XL-X | 6.19 | 6.72 | 155.25 | 0.74 |
| BiFlow-XL-V | 4.21 | 4.68 | 184.97 | **0.79** |
| BiFlow-XL-X | 4.28 | 4.68 | 183.48 | **0.79** |
| BiFlow-XL-Mix ($\tau{=}0.5$) | **4.19** | **4.65** | **185.01** | 0.79 |

Table 2 presents our main results on ImageNet. BiFlow substantially outperforms the Rectified Flow baselines across all metrics. On the L/4 model, our mixed-inference strategy (BiFlow-L-Mix) improves the FID score of Rectified Flow from 11.53 to **9.85**, a relative improvement of 14.6%. The gains are even more pronounced on the larger DiT-XL/2 model, where BiFlow-XL-Mix reduces the FID from 6.22 to **4.19**, a 32.6% relative improvement. Notably, BiFlow also achieves a significantly higher Inception Score (IS) and Precision, indicating that the generated samples are not only more realistic but also more faithful to the training data. BiFlow using the other two sampling methods also consistently outperforms the baseline, with the X-flow performing slightly better than the V-flow.

## 5.3 ABLATION STUDIES.

(a) The scale of $L_c$ versus sampling steps.

(b) Measurement of path straightness.

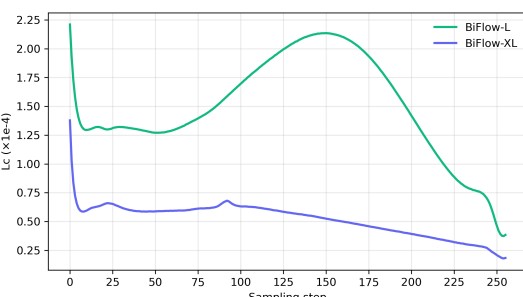

| Model | Max-dev↓ | Mean-cos↑ |
|---|---|---|
| RectifiedFlow-L | 0.256 | 0.828 |
| BiFlow-L-V | 0.250 | 0.833 |
| BiFlow-L-X | 0.248 | 0.833 |

(c) FID versus consistency weight $w_c$.

| $w_c$ | **0.0** | **0.1** | **0.2** | **0.3** |
|---|---|---|---|---|
| BiFlow-L-Mix | 11.14 | 9.85 | 9.86 | 9.91 |

Figure 2: Analysis of the effect of the consistency term in BiFlow. **(a)** The consistency loss $L_c$ measured during a 250-step ODE solver. The gap between $\mathbf{m}_\theta$ and $\tilde{\mathbf{m}}_\theta$ is largest at early steps and converges to zero. **(b)** The evaluation of path straightness shows BiFlow generates more direct paths than Rectified Flow. **(c)** A sweep over the consistency weight $w_c$ shows that $w_c > 0$ is critical.

To validate our design choices and understand how BiFlow improves the generation performance, we conduct a series of ablation studies focusing on the role of the consistency loss ($L_c$).

**The X-flow or the V-flow does not work well separately.** To isolate the effect of the X-flow training from minimizing $L_x(\theta) = \|\tilde{\mathbf{m}}_\theta(\mathbf{x}_t, t) - \mathbf{x}_1\|_2^2$, we can train a Rectified Flow model by minimizing $L_x(\theta)$ only and then draw samples from $\tilde{\mathbf{v}}_\theta$ derived from $\tilde{\mathbf{m}}_\theta$. As we mentioned before, this method is labeled as RectifiedFlow-X, and its performance is shown in Table 2. Its performance is nearly identical to the performance of the baseline Rectified Flow. Note that the V-flow $\mathbf{v}_\theta$ is trained against a corrupted image $\mathbf{x}_1 - \mathbf{x}_0$, while the X-flow is trained against a clean image $\mathbf{x}_1$. This result demonstrates that the different training targets do not bring a clear performance difference.

**The consistency loss is necessary.** In Figure 2(c), we vary the the weight $w_c$ of the consistency term $L_c(\theta)$. The best performance is achieved with $w_c \in \{0.1, 0.2\}$. When $w_c = 0$ eliminates the consistency constraint, BiFlow only has a slight performance improvement over the baseline Rectified Flow (11.14 vs. 11.53 FID). We hypothesize that the improvement is from the network sharing, which is the focus of the next experiment. This result reveals that the consistency loss $L_c$ is essential for unlocking the full potential of our method. Our qualitative study later in Figure 4 also shows that the consistency loss improves the quality of the generated images.

**The shared neural architecture brings performance gain.** Sharing the same neural architecture clearly saves computation. At the same time, they learn targets that are quite related, so sharing the architecture should also be a reasonable choice for performance consideration. To verify this hypothesis, we train two independent DiT-L models for the V-flow and the X-flow. The training objective is the same as BiFlow models above. We denote this new setup with the suffix "-2net". The performance of the

Table 3: The performance of BiFlow models using two separate networks for the V-Flow and X-Flow.

| Model | FID↓ | sFID↓ |
|---|---|---|
| BiFlow-L-V-2net | 10.92 | 12.15 |
| BiFlow-L-X-2net | 10.57 | 10.77 |

two models under this setup is shown in Table 3. The performance still improves upon the baseline Flow Matching model. The improvement is solely due to the consistency term because that's the only difference from the baseline. However, their performance cannot match that of BiFlow models with shared networks, which indicates the performance benefit of learning related targets with a single network backbone.

**Consistency helps to overcome randomness in training.**
As we analyzed in Section 3, the V-flow and the X-flow should be equivalent when there are infinite training examples, and the consistency term would not be useful. We hypothesize that the consistency between the V-flow and the X-flow is very beneficial when the model is trained with limited data and noise. To verify this hypothesis, we increase the amount of random noise examples to 4 times the training samples in *each* training batch, and therefore reduce the random noise in the training process.

Table 4: Performance of models trained with the 1:4 ratio of training samples and noise samples on CIFAR-10.

| Model | FID $\downarrow$ | sFID $\downarrow$ |
|---|---|---|
| RectifiedFlow (1:4) | 2.57 | 2.94 |
| BiFlow-Mix (1:4) | 2.28 | 2.73 |

We conducted this experiment with the CIFAR-10 dataset. The performance of the trained models is shown in Table 4. The new training strategy improves the performance of both models because it essentially increases the training batches of the both models. However, this new setup shows less performance improvement with BiFlow than the standard setup: the reduction of FID is (2.57 - 2.28) from the previous reduction (3.12 - 2.41). This provides strong evidence supporting our hypothesis.

**BiFow slightly straightens sampling paths**   We compare BiFlow and Rectified Flow in terms of the straightness of sampling paths. The straightness is measured by Max-dev (the maximum deviation from the chord connecting the two ends of the path) and Mean-cos (average cosine similarity between each step and the chord). Lower Max-dev or higher Mean-cos indicates more straight paths. The results are shown in Figure 2(b). BiFlow achieves slightly straighter paths. One possible explanation is that the X-flow aims to predict the destination and thus has a better chance to learn straight paths. We also plot the consistency loss $L_c$ in Figure 2(a) at each step of the ODE path. For both L and XL models, the difference is very small (at the scale of 1e-4), although the two models share a similar pattern. The inconsistency is highest near $t = 0$ where the path is most uncertain, and converges towards zero as $t \to 1$.

(a) FID-50K vs. Training Steps. Left: L Models. Right: XL Models.

(b) Training and Sampling Time Comparison (seconds) on ImageNet

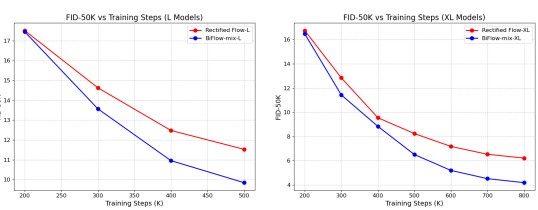

| Model | Tr./epoch | Spl./step |
|---|---|---|
| Rectified Flow-L | 0.80 | 0.09 |
| BiFlow-V-L | 1.01 | 0.10 |
| Rectified Flow-XL | 0.78 | 0.51 |
| BiFlow-V-XL | 0.81 | 0.56 |

Figure 3: Efficiency and Training Dynamics. **(a)** BiFlow has lower FID score than the baseline at every checkpoint. **(b)** The training overhead is negligible and sampling cost is identical.

**Computation efficiency.**   Figure 3 demonstrates that BiFlow's significant performance gains are achieved with high efficiency. The plots on the left show the FID-50K score as a function of training steps for both L and XL models. At every evaluation checkpoint, BiFlow maintains a consistent and substantial advantage over the standard Rectified Flow baseline, indicating not only a better final result but also faster convergence. The table on the right quantifies the computational cost. The training time per epoch for BiFlow is only marginally higher than the baseline. This is due to our efficient single-head design, which requires only one additional forward pass through the shared network backbone to compute both targets. Crucially, the sampling time per step is nearly identical, as both methods require just one model evaluation per ODE step. Taken together, these

results confirm that BiFlow is a "plug-and-play" upgrade, offering superior sample quality and faster training convergence with a negligible increase in training cost and no change to the sampling budget.

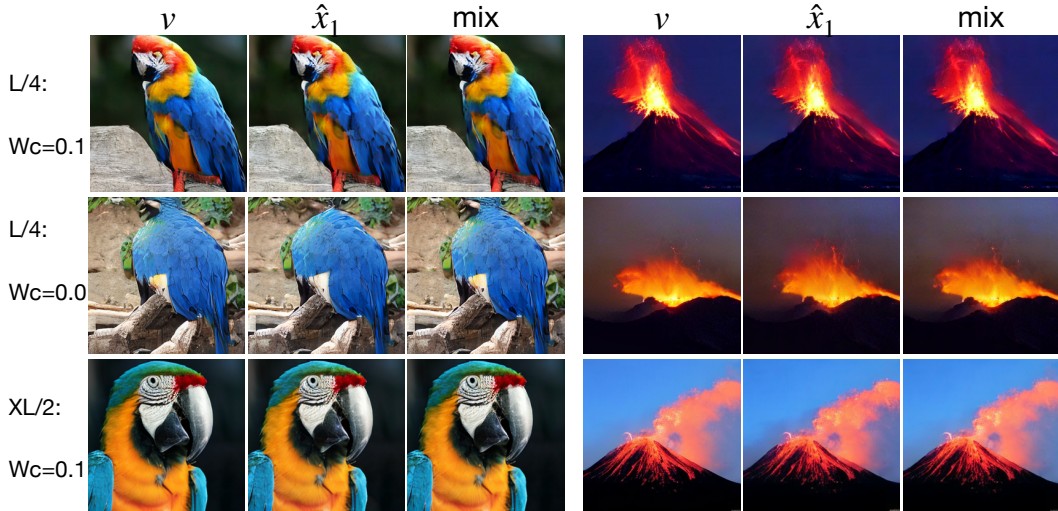

Figure 4: **Qualitative Comparison on ImageNet** $256 \times 256$. We compare samples from the L/4 and XL/2 BiFlow models using velocity-only ($v$), endpoint-only ($\hat{x}_1$), and mixed inference. The bottom row (XL/2) shows higher fidelity than the L/4 model. The middle row ablates our consistency loss ($w_c = 0.0$), revealing a clear visual divergence between the velocity and endpoint predictions. The top row ($w_c = 0.1$) demonstrates that our proposed consistency loss forces the two predictions into alignment, producing more coherent and higher-quality samples.

**Qualitative Analysis.** Figure 4 provides a qualitative analysis of BiFlow. As expected, the larger XL/2 model (bottom row) generates samples with noticeably higher fidelity and finer detail than the L/4 model (top row), consistent with the quantitative metrics. The critical comparison is between the model trained with our consistency loss ($w_c = 0.1$, top row) and the ablation without it ($w_c = 0.0$, middle row). When $w_c = 0.0$, the model's two predictions are unconstrained and learn inconsistent vector fields. This is visually apparent: the samples generated respectively from the V-flow and the X-flow exhibit clear discrepancies in texture, lighting, and fine details. For example, the parrot's perch has noticeable artifacts in the V-flow sample that are absent in the X-flow. By introducing the consistency loss, these two predictions are forced into agreement. The resulting images from both flows become nearly identical and have reduced artifacts. This visually confirms the significance of the consistent loss, which enforces the entire model to learn a coherent generation flow function.

## 6 CONCLUSION AND LIMITATIONS

**Limitations.** First, our training objective assumes a linear rectified path and ODE-based sampling. Extending it to other types of conditional paths would require some careful analysis. Second, BiFlow needs to be validated at generation tasks with high-resolution images and other modalities (e.g., text-to-image, video). Third, BiFlow slightly increases training time.

**Conclusion.** We introduced **BiFlow**, a plug-and-play upgrade to rectified flows that trains a single model to predict both instantaneous velocity and likely destination, tied together by a lightweight algebraic consistency penalty. At inference, a simple time-scheduled switch provides a sampling policy that consistently improves generation. Empirically, BiFlow improves sample quality on CIFAR-10 and ImageNet $256 \times 256$ at identical sampling cost and negligible training overhead, indicating that learning both where to go and how to move leads to stronger rectified-flow generators. The results suggest that explicitly coupling local motion with a global destination is a broadly useful principle for rectified-flow models without architectural upheaval or extra sampling compute.

**Reproducibility Statement.** We make reproducibility a priority. The full ImageNet setup (data preprocessing, frozen VAE encoder, latent scaling, and backbone details) is specified in Appendix A.1 with Tables 5 and 6. Optimization hyperparameters and training schedule are summarized in Table 8; sampling and guidance settings (including solver, step budget, and switch policy) are in Table 9. PyTorch-like pseudo-code for both training and sampling is provided in Appendix A.2 (Algorithms 1 and 2). All ablation definitions and metric protocols (FID-50K, sFID, IS, Precision) are described in the main text and figures, with straightness metrics formalized in Appendix A.

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

## A  APPENDIX

### APPENDIX A.1  IMAGENET EXPERIMENT DETAILS

This section documents the full training and evaluation setup used for ImageNet $256\times256$ with DiT-L/4 and DiT-XL/2 backbones. Unless stated, BiFlow and the Rectified-Flow baselines share identical settings.

#### DATA AND LATENT ENCODING

**Dataset and preprocessing.**    We train on ImageNet-1K (train split) with class-conditional labels. Images are center-cropped and resized to $256 \times 256$, followed by random horizontal flip and normalization to $[-1, 1]$.

**Latent space (VAE).** All models operate in the latent space of a frozen, pretrained VAE encoder (no finetuning). We use `diffusers` `AutoencoderKL.from_pretrained("stabilityai/sd-vae-ft-ema")`. Latents are scaled by the standard factor $0.18215$ (as in Stable Diffusion).

Table 5: Latent-space and encoder specifics.

| Item | Encoder | Trainable | Image Size | Downsample | Latent Size | Latent Channels |
|---|---|---|---|---|---|---|
| VAE | SD-VAE-FT-{ema\|mse} | No (frozen) | 256×256 | ×8 | 32×32 | 4 |

#### MODEL, CONDITIONING, AND MODE BIT

**Backbones.**    We use DiT-L/4 and DiT-XL/2 transformers (patch sizes 4 and 2 respectively). Exact widths/depths follow DiT; we summarize placeholders below for completeness.

Table 6: Backbone summary (filled from the provided model definitions).

| Model | Input Size | In Ch. | Patch | Tokens | Depth | Hidden | Heads | PosEnc |
|---|---|---|---|---|---|---|---|---|
| DiT-L/4 | 32 | 4 | 4×4 | 8×8=64 | 24 | 1024 | 16 | 2D sin/cos (frozen) |
| DiT-XL/2 | 32 | 4 | 2×2 | 16×16=256 | 28 | 1152 | 16 | 2D sin/cos (frozen) |

**Conditioning streams and BiFlow mode.**    We sum three embeddings into the adaLN-Zero conditioning vector at every block:

$$e(t, y, b) \;=\; E_t(t) \;+\; E_y(y) \;+\; E_b(b).$$

Time uses a sinusoidal projection (`frequency_embedding_size=` 256) followed by an MLP to the hidden size. Class labels use an embedding table of size `num_classes + 1` when classifier-free guidance (CFG) dropout is enabled. The BiFlow *mode bit* $b \in \{0, 1\}$ selects which prediction the single head should output (local motion when $b=0$, global endpoint when $b=1$).

Table 7: Embedding dimensions and fusion.

| Stream | Source | Dim | Notes | Fusion |
|---|---|---|---|---|
| Time | sinusoidal → MLP | $d_{\text{hid}}$ | freq emb size 256 | |
| Class | embedding table | $d_{\text{hid}}$ | (`num_classes`+1) if CFG | $E_t + E_y + E_b$ |
| Mode bit $b$ | embedding table (2 entries) | $d_{\text{hid}}$ | $b=0$: local motion;  $b=1$: endpoint | |

#### OPTIMIZATION AND TRAINING SCHEDULE

We train with AdamW (constant learning rate $1\times10^{-4}$, no weight decay) and exponential moving average (EMA) of weights. TensorFloat-32 (TF32) is enabled on A100/H200.

Table 8: Optimizer, precision, and regularization.

| Setting | Optimizer | LR | Weight Decay | Betas | Eps | EMA Decay | AMP |
|---|---|---|---|---|---|---|---|
| All | AdamW | $1 \times 10^{-4}$ | 0 | (0.9, 0.999) | $1 \times 10^{-8}$ | 0.9999 | False |

SAMPLING AND EVALUATION

We integrate the probability-flow ODE using Dopri5 with a fixed budget of 250 steps. For BiFlow we keep compute parity by evaluating only the active branch per step (one forward pass per step). We report FID-50K, sFID, IS, and Precision.

Table 9: Sampling configuration and guidance.

| Model | Solver | Steps | Switch $\tau$ | CFG Scale |
|---|---|---|---|---|
| DiT-L/4 | Dopri5 | 250 | 0.5 | 4.0 |
| DiT-XL/2 | Dopri5 | 250 | 0.5 | 1.5 |

PATH STRAIGHTNESS METRICS: MEAN_COS AND MAX-DEV

**Setup and notation.** For a batch of discrete sampling paths $\{x_k^{(b)}\}_{k=0}^S$ with $b = 1, \ldots, B$ and $x_k^{(b)} \in \mathbb{R}^D$ (flattened images), define the *chord*

$$c^{(b)} = x_S^{(b)} - x_0^{(b)}, \qquad u^{(b)} = \frac{c^{(b)}}{\|c^{(b)}\|} \quad \text{(unit chord).} \tag{11}$$

Let the *step vector* be

$$s_k^{(b)} = x_{k+1}^{(b)} - x_k^{(b)}, \qquad k = 0, \ldots, S - 1. \tag{12}$$

**Mean cosine to the chord (mean_cos).** The per–time-step cosine between the step and the chord direction is

$$\gamma_k^{(b)} = \frac{\langle s_k^{(b)}, u^{(b)} \rangle}{\|s_k^{(b)}\|} = \frac{\left\langle x_{k+1}^{(b)} - x_k^{(b)}, \frac{x_S^{(b)} - x_0^{(b)}}{\|x_S^{(b)} - x_0^{(b)}\|} \right\rangle}{\|x_{k+1}^{(b)} - x_k^{(b)}\|}. \tag{13}$$

We average over steps and the batch:

$$\texttt{mean\_cos} = \frac{1}{BS} \sum_{b=1}^B \sum_{k=0}^{S-1} \gamma_k^{(b)}. \tag{14}$$

Values closer to 1 indicate that steps remain well aligned with the global chord (straighter paths).

**Maximum lateral deviation (max-dev).** For each intermediate point, decompose $x_k^{(b)}$ into chord-parallel progress and orthogonal residual. Define the signed progress along the chord,

$$d_k^{(b)} = \langle x_k^{(b)} - x_0^{(b)}, u^{(b)} \rangle, \tag{15}$$

the orthogonal projection onto the chord,

$$p_k^{(b)} = x_0^{(b)} + d_k^{(b)} u^{(b)}, \tag{16}$$

and the residual (lateral) displacement

$$r_k^{(b)} = x_k^{(b)} - p_k^{(b)}, \qquad \delta_k^{(b)} = \|r_k^{(b)}\|. \tag{17}$$

The per-path maximum lateral deviation (normalized by chord length) is

$$\texttt{max-dev}^{(b)} = \frac{1}{\|c^{(b)}\|} \max_{0 \le k \le S} \delta_k^{(b)}. \tag{18}$$

We report the batch average:

$$\texttt{max-dev} = \frac{1}{B} \sum_{b=1}^{B} \texttt{max-dev}^{(b)}. \tag{19}$$

Smaller values indicate paths that hug the straight line more tightly (fewer detours).

**Implementation notes.** All vectors $x_k^{(b)}$ are flattened before inner products; norms are Euclidean. For numerical stability, denominators are clamped, e.g. $\|c^{(b)}\| \leftarrow \max(\|c^{(b)}\|, \varepsilon)$ and $\|s_k^{(b)}\| \leftarrow \max(\|s_k^{(b)}\|, \varepsilon)$ with $\varepsilon \approx 10^{-8}$.

## APPENDIX A.2    PSEUDO-CODE OF BIFLOW

---

**Algorithm 1** BiFlow Training: PyTorch-like Pseudo-code

```python
class BiFlowTrainer(nn.Module):
    def __init__(self, model, vae, transport, lr=1e-4, ema_decay=0.9999):
        super().__init__()
        self.net = model # single-head, b in {0,1}
        self.vae = vae.eval() # frozen encoder
        self.transport = transport # provides path plan, time sampling
        self.opt = torch.optim.AdamW(self.net.parameters(), lr=lr, weight_decay
            =0.0)
        self.ema = copy.deepcopy(self.net).eval()
        self.ema_decay = ema_decay
        torch.backends.cuda.matmul.allow_tf32 = True
        torch.backends.cudnn.allow_tf32 = True

    @torch.no_grad()
    def _update_ema(self):
        for p_ema, p in zip(self.ema.parameters(), self.net.parameters()):
            p_ema.mul_(self.ema_decay).add_(p, alpha=1.0 - self.ema_decay)

    def training_step(self, x_img, y, w_v=1.0, w_x=1.0, w_c=0.1, time_weight=
        False):
        with torch.no_grad():
            z = self.vae.encode(x_img).latent_dist.sample().mul_(0.18215) # (B
                ,4,32,32)

        # sample t, x0, x1, and path (xt, ut) from transport
        t, x0, x1 = self.transport.sample(z) # shapes match z
        t, xt, ut = self.transport.path_sampler.plan(t, x0, x1)

        b0 = torch.zeros(xt.size(0), device=xt.device, dtype=torch.long) #
            velocity branch
        b1 = torch.ones (xt.size(0), device=xt.device, dtype=torch.long) #
            endpoint branch
        v_pred = self.net(xt, t, y=y, b=b0) # local motion
        x1_pred = self.net(xt, t, y=y, b=b1) # global endpoint

        L_v = ((v_pred - ut) ** 2).mean(dim=(1,2,3)).mean()
        if time_weight:
            tw = 0.5 + t.view(-1,1,1,1)
            L_x1 = (tw * (x1_pred - x1) ** 2).mean(dim=(1,2,3)).mean()
        else:
            L_x1 = ((x1_pred - x1) ** 2).mean(dim=(1,2,3)).mean()

        t_exp = t.view(-1,1,1,1)
        x1_from_v = xt + (1 - t_exp) * v_pred
        L_c = ((x1_pred - x1_from_v) ** 2).mean(dim=(1,2,3)).mean()

        loss = w_v * L_v + w_x * L_x1 + w_c * L_c
        return loss, {'L_v': L_v, 'L_x1': L_x1, 'L_c': L_c}

    def train_loop(self, loader, epochs, log_every=100):
        self.net.train()
        for ep in range(epochs):
            for i, (x_img, y) in enumerate(loader):
                x_img, y = x_img.cuda(non_blocking=True), y.cuda(non_blocking=
                    True)
                loss, logs = self.training_step(x_img, y)
                self.opt.zero_grad(set_to_none=True)
                loss.backward()
                self.opt.step()
                self._update_ema()
                if (i + 1) % log_every == 0:
                    _ = (loss.item(), logs['L_v'].item(), logs['L_x1'].item(),
                        logs['L_c'].item())
```

---

918
919
920
921
922
923
924
925

**Algorithm 2** BiFlow-mix Sampling: PyTorch-like Pseudo-code

```
class BiFlowSampler:
    def __init__(self, model, vae, transport, cfg_scale=1.0, tau=0.5):
        self.net = model.eval()
        self.vae = vae.eval() # frozen decoder (optional for image output)
        self.transport = transport
        self.cfg_scale = cfg_scale
        self.tau = tau # switch between endpoint-induced vs direct motion

    @torch.no_grad()
    def _drift(self, x, t, y):
        if t[0] <= self.tau:
            b = torch.ones(x.size(0), device=x.device, dtype=torch.long) #
                endpoint branch
            x1_pred = self.net(x, t, y=y, b=b)
            t_safe = torch.clamp(1 - t, min=1e-3)
            v = (x1_pred - x) / path.expand_t_like_x(t_safe, x) # endpoint-
                induced motion
        else:
            b = torch.zeros(x.size(0), device=x.device, dtype=torch.long) #
                velocity branch
            v = self.net(x, t, y=y, b=b)
        return v

    @torch.no_grad()
    def _drift_cfg(self, x, t, y):
        if self.cfg_scale <= 1.0:
            return self._drift(x, t, y)
        half = x[: len(x)//2]
        xin = torch.cat([half, half], dim=0)
        vin = self._drift(xin, t, y)
        eps, rest = vin[:, :3], vin[:, 3:]
        cond, uncond = torch.chunk(eps, 2, dim=0)
        guided = torch.cat([uncond + self.cfg_scale * (cond - uncond)]*2, dim=0)

        return torch.cat([guided, rest], dim=1)

    @torch.no_grad()
    def sample(self, B, y, steps=250, method='euler'):
        z = torch.randn(B, 4, 32, 32, device=y.device)
        t0, t1 = self.transport.check_interval(self.transport.train_eps, self.
            transport.sample_eps, sde=False, eval=True)
        if method == 'euler':
            dt = (t1 - t0) / steps
            for k in range(steps):
                t_scalar = t0 + k * dt
                t = torch.full((B,), t_scalar, device=z.device)
                v = self._drift_cfg(z, t, y)
                z = z + v * dt
        else:
            solver = ODESolver(self._drift_cfg, t0=t0, t1=t1, steps=steps) #
                placeholder
            z = solver.integrate(z, y)
        x = self.vae.decode(z / 0.18215) # optional decode
        return x, z
```

965
966
967
968
969
970
971

## A.3 ADDITIONAL VISUAL RESULTS FOR CIFAR-10

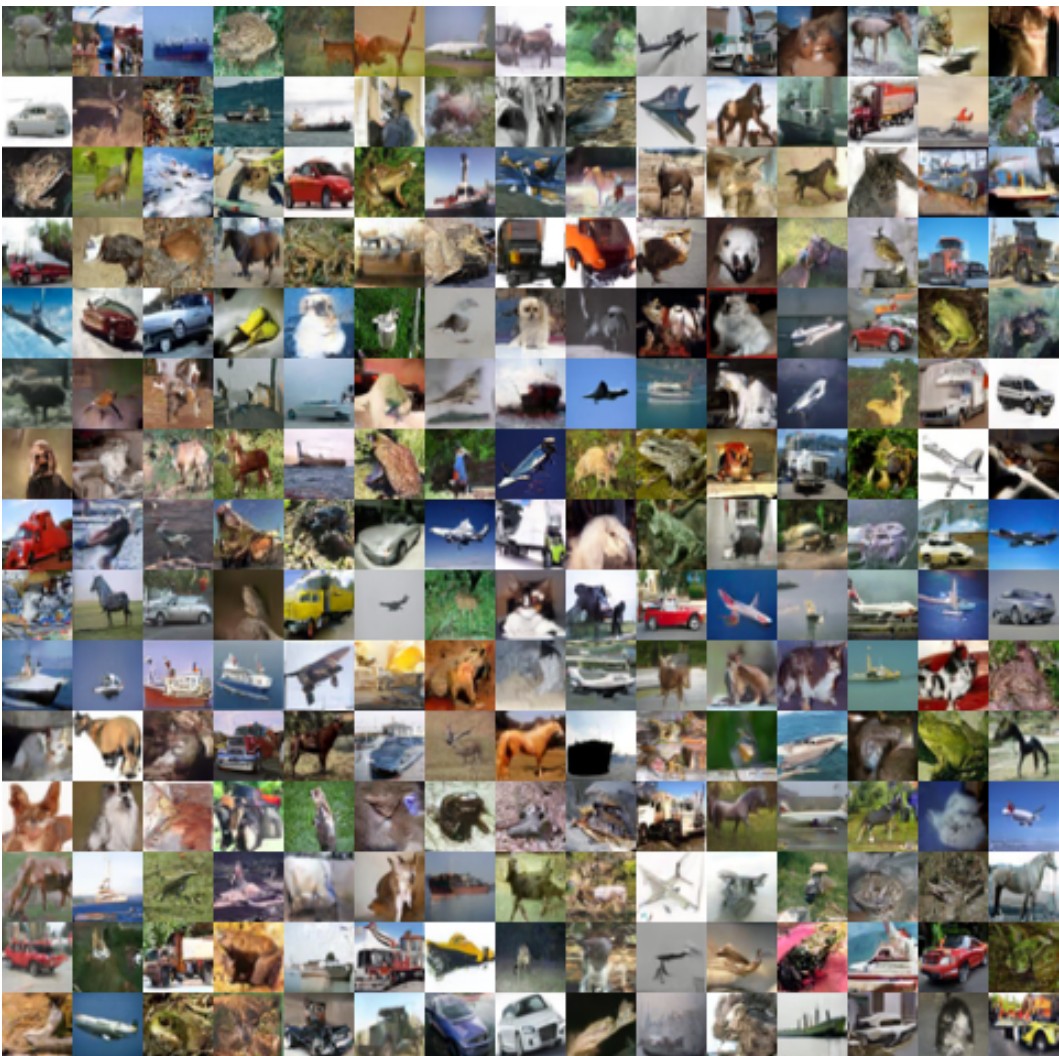

Figure 5: Samples for CIFAR-10.

## A.4 ADDITIONAL VISUAL RESULTS FOR IMAGENET

All samples are from BiFlow-mix-XL, with cfg=4.0.

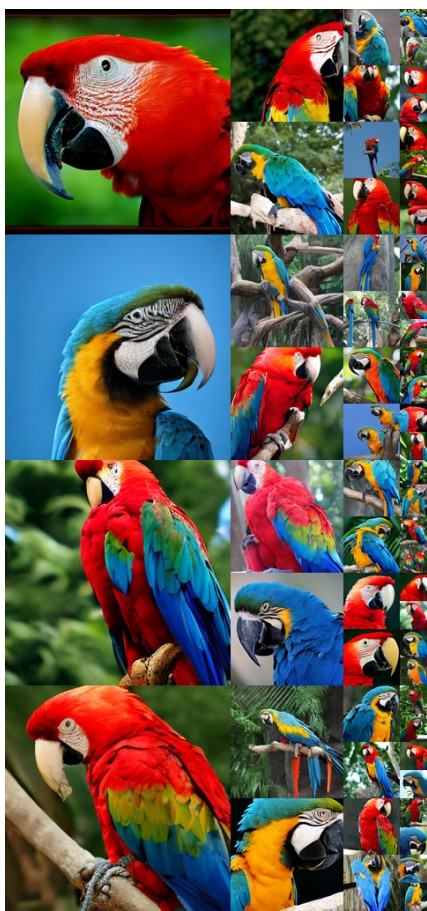

Figure 6: Samples for ImageNet class 88 (macaw).

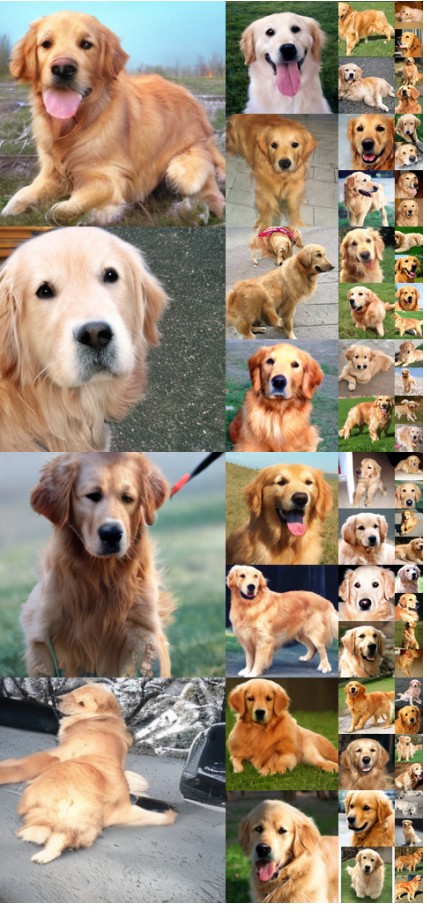

Figure 7: Samples for ImageNet class 207 (golden retriever).

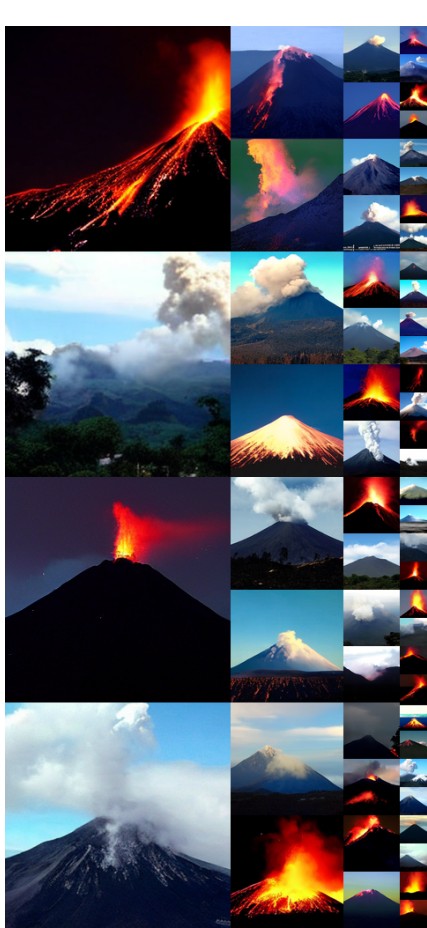

Figure 8: Samples for ImageNet class 980 (volcano).

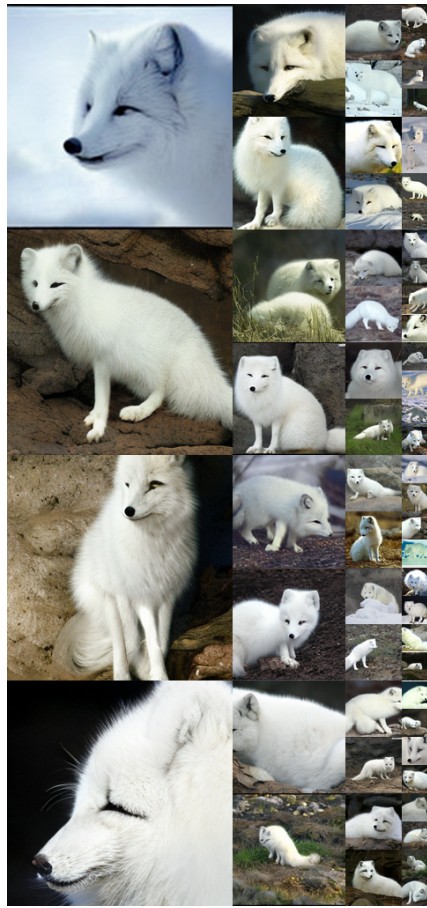

Figure 9: Samples for ImageNet class 279 (arctic fox).

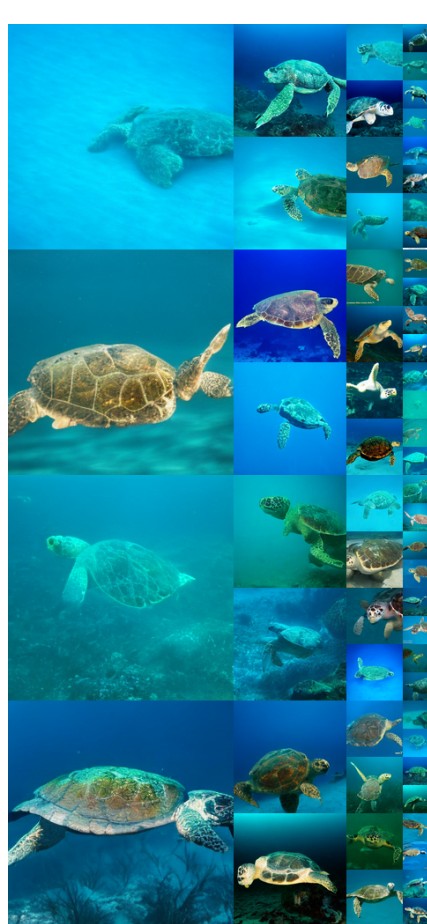

Figure 10: Samples for ImageNet class 33 (loggerhead turtle).

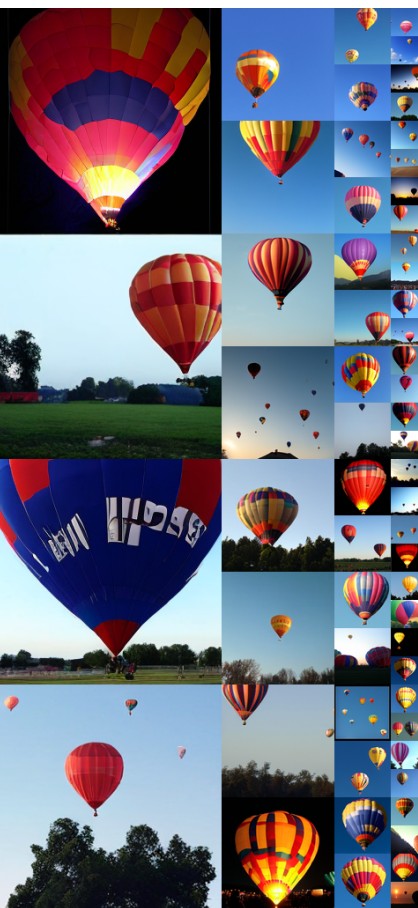

Figure 11: Samples for ImageNet class 417 (balloon).

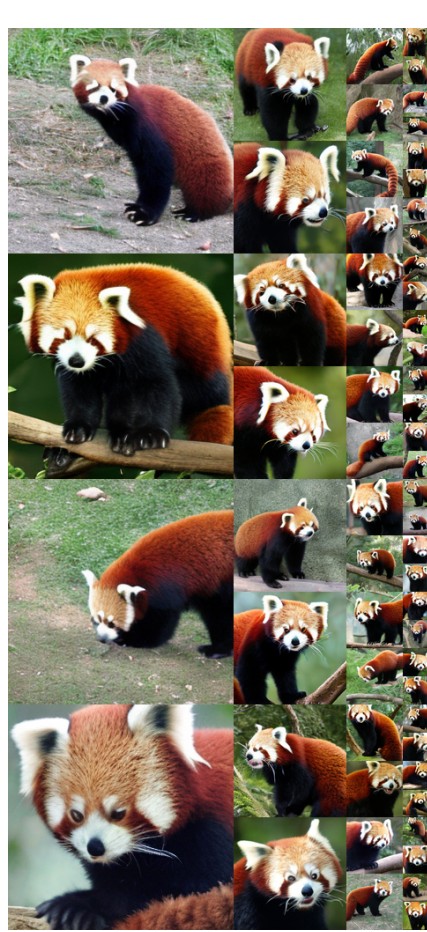

Figure 12: Samples for ImageNet class 387 (red panda).

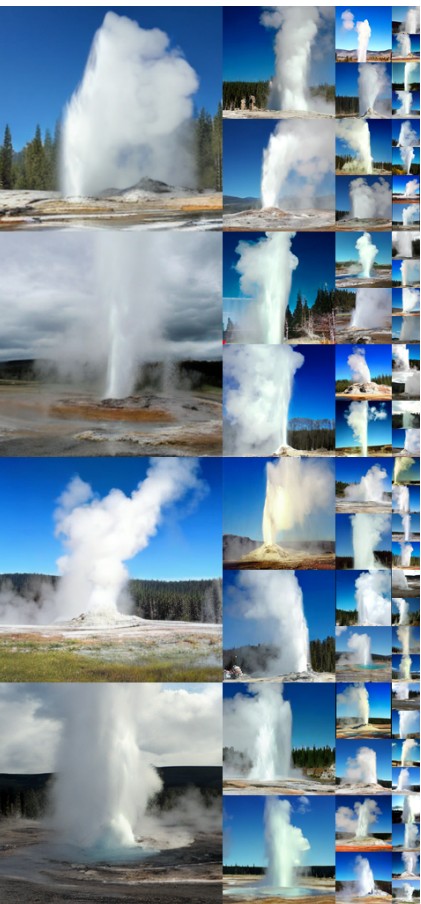

Figure 13: Samples for ImageNet class 974 (geyser).

