# OpenReview forum: "A Consistent Flow Model Learning Both Where to Go and How to Move"
_ICLR.cc/2026/Conference — Submitted to ICLR 2026_

### Official Review · Reviewer_1MXW · 2025-11-01

**Soundness:** 2
**Presentation:** 3
**Contribution:** 2
**Rating:** 4
**Confidence:** 3

**Summary:**

This paper proposes BiFlow, a rectified‑flow variant that trains a single backbone to predict both the instantaneous velocity and the endpoint, coupled by a lightweight consistency loss. In inference, BiFlow selects the drift from the velocity or the endpoint‑induced field via a time‑scheduled switch. On CIFAR‑10 and ImageNet‑256 with UNet/DiT backbones, BiFlow reports improved FID/IS/Precision and slightly straighter ODE paths; ablations indicate the consistency term helps. The paper claims a near‑identical sampling cost to baseline and “theoretically‑grounded” switching.

**Strengths:**

1. A single shared backbone with a binary mode flag outputs velocity or endpoint; a consistency penalty ties them together. Clear training/sampling diagrams aid understanding.

2. The paper examines consistency weight, single-goal variant, and shared versus two-network training, demonstrating that the consistency term and shared trunk make significant contributions to performance and path straightness.

3. There are empirical gains on standard setups. Reasonable reproducibility details are provided.

**Weaknesses:**

1. Sampling‑compute inconsistency. The main text claims identical per‑step cost, yet Algorithm 2 evaluates both heads each step before mixing, implying two forward passes; the Appendix code uses a one‑branch switch. Compute parity and timing claims need correction and reconciliation.

2. Unclear CFG pseudo‑code. The sampler splits channels (vin[:, :3] in Algorithm 2) and lacks a proper unconditional path, which is incompatible with rectified‑flow parameterization in latent space (4 channels). Guidance must combine in the drift space with a true null label.

3. Limited theory analysis & baselines. The switch is called “theoretically‑grounded,” but the analysis is not enough; related dual-objective/consistency methods are not compared under matched budgets, leaving the novelty and significance of the switch under-substantiated.

4. Equation and table inconsistencies. Eq. (3) implies a 1/(1-t)^2 factor, but Eq. (5) says it “removes 1/t^2” and drops the weighting without analysis, changing the optimum and confusing readers. Table 1 reports BiFlow‑Mix IS = 0.65, very different from neighboring values (~9.6). All numbers should be rechecked for accuracy.

**Questions:**

1. What formal criterion supports the time‑switch policy (e.g., bias/variance of V‑ vs X‑flow as a function of t) and the choice (tau=0.5)? Could the authors provide theory to support or targeted ablations?

2. For V/X/Mix, exactly how many forward passes per ODE step were used, and what are the FLOPs/wall‑clock per sample? Reconcile Algorithm 2 with Appendix A.2.

3. If the endpoint loss omits 1/(1-t)^2, what distribution over t preserves equivalence? Otherwise, why does unweighted regression not bias $m_theta$?

4. How is classifier‑free guidance implemented for flows? What is the unconditional label, and in which space are predictions combined during guidance?

5. Are all FID/IS/Precision computed on decoded RGB with the same frozen VAE for all methods? Please clarify the metric domain and any VAE confounds.

---

### Official Review · Reviewer_WLd1 · 2025-11-01

**Soundness:** 2
**Presentation:** 2
**Contribution:** 2
**Rating:** 4
**Confidence:** 3

**Summary:**

This paper proposed BiFlow that predicts the velocity and the target at the same time. Combining the two regression targets, this method can stabilize optimization and improve the straightness of generation paths.

**Strengths:**

- the paper is well-written and easy to follow
- the method and motivation is clear

**Weaknesses:**

- only experiments on ImageNet 256 is included, which is not enough
- predicting v and x at the same time might cause conflict with the same backbone, especially in more complicated text-to-image applications. If larger heads are needed for those scenarios, the inference overhead might be considerable.
- the performance on ImageNet256 is far from SOTA

**Questions:**

How about applying the method to text-to-image?

---

### Official Review · Reviewer_PBBY · 2025-11-03

**Soundness:** 3
**Presentation:** 4
**Contribution:** 1
**Rating:** 2
**Confidence:** 5

**Summary:**

The paper tackles the task of improving flow-based generative models. The authors propose training a standard Flow Matching model with both a v-prediction and x-prediction target simultaneously. This is achieved by adding a binary conditioning flag to the input that specifies which target type the model should predict. The network is then trained using a combination of two standard regression losses (one for each target type) along with a consistency loss between them. The method is evaluated on CIFAR10 and ImageNet256, showing improvements over the baseline Flow Matching model.

**Strengths:**

- The main idea of the paper is very simple and straightforward.
- The paper is very well written and easy to follow.
- The experiments and ablations are fairly extensive.

**Weaknesses:**

- The paper has extremely limited novelty. The idea of training a diffusion or flow-based model with a combination of $\epsilon$-, $x$-, and $v$-prediction targets is not new. It is also well known that these parameterizations are effectively equivalent in their training signals and can related through a weighting function $w(t)$ in the loss. Furthermore, as noted in the related works section, this idea has already been applied to diffusion models, which are essentially identical to Flow Matching when using a Gaussian prior and independent coupling.
- The experimental results are significantly below the current state of the art, with ImageNet-256 FID scores of 9.85 compared to under 1.5 for leading models. This raises concerns about how well the baselines were tuned and whether similar gains could have been achieved through better hyperparameter choices.
- Following the previous point, Table 4 shows FID values obtained with roughly four times the batch size used in the main experiments. This change substantially narrows the gap between the baseline and BiFlow, suggesting that the experimental setup may not have been fully optimized.
- Some training details, such as the batch size, are missing from the appendix.

**Questions:**

- For the model architecture, why not follow the cited prior work [1] and add multiple output heads to the final layer of the network? This would allow obtaining both types of outputs in a single forward pass, rather than requiring two separate passes as in the current implementation.
- Have the authors considered applying this approach to more optimized network architectures such as EDM or EDM2?

 [1] Benny, Yaniv, and Lior Wolf. "Dynamic dual-output diffusion models." Proceedings of the IEEE/CVF Conference on Computer Vision and Pattern Recognition. 2022.

---

### Meta-Review · Area_Chair_2Yux · 2026-01-07

**Summary:**

The authors did not submit the rebuttal. The reviewers were unconvinced on the positive side, with one decision with reject and two decisions with marginally below the acceptance threshold. They agreed that this work requires additional effort to meet the acceptance bar of ICLR. Thus, I am inclined not to accept this draft at this stage. Thank you for your effort. It is an interesting work. I hope the input from the reviewers will help you further improve this work.

**Reviewer Concerns:**

The submission has limited novelty and contributions.

**Reviewer Scores:**

Rating: 2: reject, not good enough
Rating: 4: marginally below the acceptance threshold. But would not mind if paper is accepted
Rating: 4: marginally below the acceptance threshold. But would not mind if paper is accepted

---

### Decision · Program_Chairs · 2026-01-26

Reject